# Clinical Predictors of 3- and 6-Month Outcome for Mild Traumatic Brain Injury Patients with a Negative Head CT Scan in the Emergency Department: A TRACK-TBI Pilot Study

**DOI:** 10.3390/brainsci10050269

**Published:** 2020-05-01

**Authors:** Debbie Y. Madhok, John K. Yue, Xiaoying Sun, Catherine G. Suen, Nathan A. Coss, Sonia Jain, Geoffrey T. Manley

**Affiliations:** 1Department of Emergency Medicine and Neurology, University of California San Francisco, Suite 6A, 1001 Potrero Ave, Building 5, Suite 6A, San Francisco, CA 94110, USA; 2Department of Neurological Surgery, University of California San Francisco, 505 Parnassus Ave, Rm M779, San Francisco, CA 94143, USA; john.yue@ucsf.edu (J.K.Y.); Nathan.Coss@ucsf.edu (N.A.C.); ManleyG@ucsf.edu (G.T.M.); 3Brain and Spinal Injury Center, Zuckerberg San Francisco General Hospital, 1001 Potrero Ave, Bldg. 1, Rm 101, San Francisco, CA 94110, USA; catherine.suen@hsc.utah.edu; 4Department of Family Medicine and Public Health, University of California San Diego, 9500 Gilman Drive, San Diego, CA 92093, USA; xisun@ucsd.edu (X.S.); sojain@ucsd.edu (S.J.); 5Department of Neurology, University of Utah School of Medicine, 175 North Medical Drive East, Salt Lake City, UT 84132, USA

**Keywords:** mild traumatic brain injury, emergency department, clinical predictors, outcome

## Abstract

A considerable subset of mild traumatic brain injury (mTBI) patients fail to return to baseline functional status at or beyond 3 months postinjury. Identifying at-risk patients for poor outcome in the emergency department (ED) may improve surveillance strategies and referral to care. Subjects with mTBI (Glasgow Coma Scale 13–15) and negative ED initial head CT < 24 h of injury, completing 3- or 6-month functional outcome (Glasgow Outcome Scale-Extended; GOSE), were extracted from the prospective, multicenter Transforming Research and Clinical Knowledge in Traumatic Brain Injury (TRACK-TBI) Pilot study. Outcomes were dichotomized to full recovery (GOSE = 8) vs. functional deficits (GOSE < 8). Univariate predictors with *p* < 0.10 were considered for multivariable regression. Adjusted odds ratios (AOR) were reported for outcome predictors. Significance was assessed at *p* < 0.05. Subjects who completed GOSE at 3- and 6-month were 211 (GOSE < 8: 60%) and 185 (GOSE < 8: 65%). Risk factors for 6-month GOSE < 8 included less education (AOR = 0.85 per-year increase, 95% CI: (0.74–0.98)), prior psychiatric history (AOR = 3.75 (1.73–8.12)), Asian/minority race (American Indian/Alaskan/Hawaiian/Pacific Islander) (AOR = 23.99 (2.93–196.84)), and Hispanic ethnicity (AOR = 3.48 (1.29–9.37)). Risk factors for 3-month GOSE < 8 were similar with the addition of injury by assault predicting poorer outcome (AOR = 3.53 (1.17–10.63)). In mTBI patients seen in urban trauma center EDs with negative CT, education, injury by assault, Asian/minority race, and prior psychiatric history emerged as risk factors for prolonged disability.

## 1. Introduction

In 2014, at least 2.5 million people were treated for traumatic brain injury (TBI) in U.S. emergency departments (ED), of which 80%–90% were classified as “mild” (mTBI) with a Glasgow Coma Scale (GCS) score of 13–15 [1]. The frequency of mTBI-related emergency department visits increased from 569.4/100,000 persons in 2006 to 807.9/100,000 in 2012 and continues at an increase of 7% annually [2]. While all-cause mortality for mTBI is low (1.4%) [3], patients are at significant risk of psychiatric sequelae [4] and cognitive impairment postinjury [5]. It is estimated that 20%–70% of mTBI patients suffer from persistent functional, cognitive and/or neuropsychological symptoms at or beyond 3 months postinjury [6,7]. mTBI patients often suffer disruptions to functional and social wellbeing postinjury and require greater use of healthcare resources [8]—upwards of $17 billion annually in the U.S. [9]. However, indications for referral to these necessary resources remain poorly understood, masking the true burden of mTBI. In a recent large study of over 1.1 million adult patients who suffered TBI in the state of California from 2005–2014, 40.5% had a revisit during the first year, and, of those revisiting patients, 13.8% had five or more revisits within that first year [10]. A study of 2787 mTBI patients showed that 5% returned unexpectedly to the ED within 72 h, with post-concussion symptoms or otherwise needing repeat clinical evaluation [11]. Identifying the predictors of negative outcomes for patients with mTBI and using this information to inform ED management strategies could improve patient outcomes and decrease ED utilization.

The evidence-driven management of mTBI is limited compared to moderate and severe TBI, in which additional prognostic factors have been identified due to their association with the need for neurosurgical intervention [12]. The heterogeneous nature of mTBI and the need for guidelines in its evaluation were demonstrated in the Transforming Research and Clinical Knowledge in Traumatic Brain Injury Pilot (TRACK-TBI Pilot) study [13], which identified subgroups within the mTBI population with divergent functional outcomes, persistence of symptoms, and neuropsychiatric impairment at 90 and 180 days postinjury [14]. Current ED guidelines support the utilization of GCS and pupillary reactivity to assess and monitor severity in head injury, and computed tomography (CT) as first-line imaging; however, these assessments are often normal or negative in mTBI patients and may not directly correlate with post-traumatic complaints [15,16,17]. Data on additional clinical factors, such as loss of consciousness (LOC), post traumatic amnesia (PTA), hypoxia, hypotension, and the use of anticoagulants and antiplatelets, are often collected in the ED, though their clinical significance is not well defined [9,18]. Variables not yet described in guidelines due to minimal literature but found to be predictive of impaired outcomes following mTBI include education level and pre-injury psychiatric history, such as anxiety and depression [4,16,19,20].

With the increasing number of mTBI patients presenting to the ED and growing evidence of the presence of persistent post-traumatic symptomatology and disability, it is critical to identify risk factors for adverse outcomes, update ED assessment and treatment guidelines, and improve the efficiency of ED utilization. This is necessary to determine the appropriate level of care and services for this heterogeneous group of individuals. In the current ED mTBI management paradigm, patients who have a CT scan without evidence of intracranial abnormalities are routinely discharged from the ED with return precautions. However, it has been found that only 25% return to follow-up, of which one-third had not completely resumed pre-injury activities [21]. Thus, beyond improving outcome, identifying mTBI patients with negative CT who are at risk for poorer outcomes can improve ED evaluation and triage paradigms, as well as healthcare utilization. The current analysis of the TRACK-TBI Pilot dataset sought to identify a cohort of patients in the ED with mTBI who are at greater risk for adverse outcomes despite an initial negative head CT scan, and assess whether traditional clinical factors measured in the ED in this patient population are adequate for predicting outcome at 3 and 6 months postinjury.

## 2. Materials and Methods

### 2.1. Study Design of the TRACK-TBI Pilot Study

The TRACK-TBI Pilot study was conducted at three U.S. Level I trauma centers (Zuckerberg San Francisco General Hospital (San Francisco, CA, USA), University of Pittsburgh Medical Center (Pittsburgh, PA, USA), and University Medical Center Brackenridge (Austin, TX, USA)) using the National Institute of Neurological Disorders and Stroke (NINDS) TBI Common Data Elements (CDEs) [13,22,23,24,25]. The TRACK-TBI Pilot inclusion criteria were external force trauma to the head, presentation to one of the three trauma centers, and a clinically indicated head CT scan, as determined by the treating emergency physician, obtained within 24 h of injury. Exclusion criteria were pregnancy, ongoing life-threatening disease (e.g., end-stage malignancy), police custody, involuntary psychiatric hold, and inability to speak English due to limitations in participation with outcome assessments.

Eligible subjects were enrolled prospectively by trained clinical research coordinators via convenience sampling from years 2010 to 2012. Institutional Review Board approval was obtained at the participating sites. Informed consent was obtained prior to study enrollment. For subjects unable to provide consent due to their injury, consent was obtained from their legally authorized representatives. Subjects were then re-consented, if cognitively able, during the course of their clinical care and/or follow-up visits for continued participation in the study.

### 2.2. Participants Included in the Current Analysis

As the goal of the current analysis was to evaluate associations between demographic, clinical and injury factors routinely collected upon ED admission for mTBI and 3- and 6-month global functional outcome, subjects from the TRACK-TBI Pilot were retrospectively included if they were aged ≥18 years with an ED admission GCS score of 13–15, without evidence of intracranial injury on ED admission head CT as determined by a blinded, central board-certified neuroradiologist, and completed 3- or 6-month Glasgow Outcome Scale-Extended (GOSE) assessment. All study personnel conducting outcome assessments were trained in-person by a certified neuropsychology coordinator prior to study commencement. The GOSE was completed by patient interview in all cases with the exception of patients physically and/or cognitively debilitated to the extent where they were unable to reliably respond to the assessment (e.g., GOSE 3 or 4, requiring assistance with activities of daily living) in which case the designated caretaker was interviewed. Candidate predictors were included based on their utility as described in large prior studies in mTBI [26,27], as well as capacity to be routinely collected by ED personnel for patients with mTBI. Demographic factors, such as race, ethnicity and education level, and clinical factors, such as prior psychiatric history, were self reported.

### 2.3. Outcome Measure

The GOSE is a measure of functional disability covering cognition, independence, employability, and social/community participation. It is assessed via structured interview and has been widely used as a standard outcome measure for TBI [28,29]. Subjects are assigned one of eight categories: 1 = dead, 2 = vegetative state, 3 = lower severe disability, 4 = upper severe disability, 5 = lower moderate disability, 6 = upper moderate disability, 7 = lower good recovery, and 8 = upper good recovery (return to baseline function). The GOSE is anchored to baseline functional status; for instance, patients who returned to their baseline level of physical, mental health and/or socioeconomic status were scored as GOSE = 8. Patients with additional physical, cognitive, functional and/or social integration burden after TBI were scored as GOSE < 8, accordingly.

For the current analysis, 3- and 6-month GOSE outcomes were dichotomized to full return to baseline function (GOSE = 8) versus presence of functional deficits (GOSE < 8). By definition, a GOSE of 7 implies deficits in resumption of social activities, problems with social integration, and persistent postconcussive symptoms resulting in functional impairment compared to baseline. Recent studies have suggested that there are distinct neuropsychological differences between the GOSE 7 and GOSE 8 groups, more specifically the prominence of neurocognitive impairment in the GOSE 7 group, suggesting that “good recovery” for GOSE 7 patients may be misleading especially after mTBI [30].

### 2.4. Statistical Analysis

Descriptive statistics are presented as medians and interquartile range (IQR) for continuous variables and proportions for categorical variables. Group differences were assessed using the Wilcoxon Rank Sum test for continuous variables and Fisher’s exact test for categorical variables. Univariate predictors with *p* < 0.10 were included in multivariable regression models, with the exception of those with cell counts of 0 on univariate analyses due to limitations on model performance. AOR for multivariable logistic regressions to predict 3- and 6-month functional deficits, and their associated 95% confidence intervals (CI), were reported for each predictor in the regression analyses. Significance was assessed at *p* < 0.05. All analyses were performed using R [31] version 3.5.1.

## 3. Results

Overall, there were 271 eligible subjects with GCS 13–15 without evidence of intracranial injury on emergency department head CT. The majority of patients were male (69.4%) and white (73.4%). The median age was 39 years, IQR ranging from 26 to 52 years. The GCS distribution of these patients was: 15 (74.9%), 14 (22.1%), 13 (3.0%). The median years of education was 14 years (IQR: 12–16 years). Psychiatric history was noted as present for 37.3% of the patients. Nearly all patients (99.6%) had sustained a closed TBI. Motor vehicle accidents accounted for 35.8% of total injuries, falls accounted for 27.3%, bicycle accidents accounted for 20.7%, and assaults accounted for 18.1% of injuries. LOC was present in 51.3% of patients and PTA was present in 49.4% of patients. The prevalence of ED hypoxia and ED hypotension was 4.8% and 1.1%, respectively. Of the 271 eligible mTBIs, 211 completed 3-month outcomes and 185 completed 6-month outcomes. Subjects who completed the outcome assessments were similar in baseline characteristics to those who missed the assessments at 3 or 6 month postinjury. At 3-months, 60.2% of subjects (n = 127) and at 6-months, 65.4% (n = 121) had functional deficits (GOSE < 8). The flowchart of included subjects is presented in Figure 1.

### 3.1. Univariate Predictors of 3- and 6-Month GOSE

Based on univariate analysis (Table 1), at the 3-month outcome, years of education were significantly lower in subjects with functional deficits (GOSE < 8) compared with those without functional deficits (GOSE = 8) (median 13 vs. 15, *p* < 0.01). A greater proportion of subjects with a baseline psychiatric history (e.g., anxiety, depression, post-traumatic stress, bipolar disorder, schizophrenia, sleep disorders, and others) had functional deficits compared with those without a psychiatric history (72% vs. 53%, *p* < 0.01). Individuals who suffered from assault as the mechanism of injury were more likely to have functional deficits than those with injuries from falls or accidents (81% vs. 56%, *p* = 0.01). A greater proportion of Asian (86%), American Indian or Alaskan (100%), and Hawaiian or Pacific Islander (100%) participants had functional deficits, compared to Caucasian participants (54%, *p* = 0.015). A nonsignificant statistical trend was observed for subjects with pre-hospital hypoxia more likely to have functional deficits compared with subjects without pre-hospital hypoxia (100% vs. 58%, *p* = 0.08). These results persist and remain significant at the 6-month GOSE measurement (Table 1) with the addition that Hispanic subjects had greater functional deficits, compared with non-Hispanic subjects (81% vs. 61%, *p* = 0.03).

No statistical associations were observed of functional deficits on 3- or 6-month GOSE < 8 vs. GOSE = 8 with gender, GCS, anticoagulant use, LOC, PTA, pre-hospital hypotension, urine toxicology screen, blood alcohol level, ED hypotension, or ED hypoxia.

### 3.2. Multivariable Predictors of 3- and 6-Month GOSE

On multivariable analysis at 3 months (Table 2), significant risk factors associated with worse outcome included fewer years of education (AOR = 0.88 per year increase in education, 95% CI: 0.78–0.998, *p* = 0.046), the presense of psychiatric history (AOR 2.30, 95% CI: 1.18–4.49, *p* = 0.015), being Asian, American Indian or Alaskan, Hawaiian, or Pacific Islander (AOR = 12.40, 95% CI: 2.66–57.77, *p* = 0.006). Assault as the mechanism of injury was also associated with an increased odds of functional deficit (AOR = 3.53, 95% CI: 1.17–10.63, *p* = 0.025).

Risk factors for functional deficits at 6 months postinjury were similar (Table 2). Lower education and the presense of psychatric history were associated with higher odds of worse outcome. Hispanic ethnicity was also a predictor for 6-month deficits (AOR = 3.48, 95% CI: 1.29–9.37, *p* = 0.01).

## 4. Discussion

This study is one of very few to examine the association of baseline clinical factors with postinjury functional outcomes in a cohort of mTBI patients presenting to EDs of Level 1 trauma centers with a negative head CT. Education level, pre-injury psychiatric history, assault as the mechanism of injury, Asian race and other minorities (American Indian or Alaskan, Hawaiian or Pacific Islander), and Hispanic ethnicity were predictive of heightened risk for functional deficits at three or six months postinjury [4,6,20,26,32,33,34,35,36,37,38]. Although there is a pervasive belief that all TBI will get better with time [39,40]. We found that the proportion of participants with functional deficits did not decrease from 3 to 6 months in patients who returned for follow-up. Therefore, modification of existing ED assessment and treatment guidelines to highlight the needs of these patients and provide appropriate referral to follow-up at earlier time points could prove beneficial.

Our finding that over 60% of patients with CT-negative, GCS 13–15 TBI have functional deficits at 3 and 6 months aligns with findings over the past decade that the proportion of deficits in mTBI patients is higher than traditionally appreciated [7,41]. Notably, commonly cited risk factors for poor outcome in severe TBI such as pre-hospital hypoxia and hypotension [42,43,44,45], were rarely seen in this cohort of patients who inherently have “milder” injuries, and factors used in ED guidelines such as LOC and PTA [46] were not associated with outcome. However, a negative CT with a GCS of 13–15 does not necessarily portend that the patient will recover to baseline after a short period of cognitive rest—in fact, a subset of these patients likely require increased surveillance and strict return to activity precautions. Our finding that GOSE deficits associated with a pre-injury history of psychiatric conditions were maintained at three and six months suggests that longer follow-up both in clinical settings and in research design may be critical in order to refer for treatment and obtain accurate outcome assessments, respectively.

Despite the evidence of persistent symptoms months to years postinjury [47,48,49,50] a large proportion of mTBI patients do not receive follow-up care. In a study of 831 mTBI patients, only 42% received TBI educational materials at ED discharge [51] despite evidence that patient educational information and targeted follow-up have been shown to reduce post-concussive symptoms [52,53,54,55]. In this same cohort, nearly 40% of CT-positive, mTBI patients had not seen a physician by three months following injury [51]. Even in those with persistent symptoms, only 41% were seen by a practitioner by two weeks and 44% by three months [51].

The ED physician is uniquely positioned to stratify patients that are in greater need of support and resources given that the majority of the triage, work-up, and injury education of mTBIs occur in the ED, and the vast majority of mTBI patients are discharged directly from the ED. Currently, patient education level and pre-injury psychiatric history information are not routinely collected in the ED. Mechanism of injury, race, and ethnicity are frequently collected in the ED; however, these factors rarely inform decision making after TBI. Including the assessment of these evidence-based clinical factors in the guidelines for ED management of mTBI would trigger appropriate referral for the patients most likely to develop persistent sequelae and demonstrate the justification for more accessible services.

Resources such as social work, follow-up referrals, and patient education regarding mTBI are sparse compared with those available for moderate and severe TBI [56,57]. One potential method to improve follow-up is to ensure ED social workers discuss follow-up with patients prior to discharge [58]. While treatment for mTBI remains variable, evidence has shown that cognitive rehabilitation, psychosocial support, and consistent follow-up with specialists can reduce symptomatology, decrease the time needed to return to work or baseline social integration, and improve the quality of life [59,60,61,62]. Utilizing existing infrastructure, such as patients’ primary care physicians, to improve follow-up when the brain is most receptive to rewiring [63,64], could help alleviate both post-concussive symptoms and observed long-term effects, such as neurocognitive decline [65,66].

### Limitations

We recognize several limitations to our analysis. We analyzed mTBI patients from a convenience sample that may not be representative of the entire mTBI population. We also recruited from U.S. Level I trauma centers, which predominantly care for patients with more severe injuries and a greater likelihood of comorbidities. We focused on investigating variables routinely collected as part of ED admission that may be of value in overall functional recovery, and did not analyze for trajectories of recovery, which would be a prime topic for future investigations. The number of participants in non-White racial groups is small. Variables such as pre-hospital hypoxia also suffered from low cell counts on univariate analyses and were not included in multivariable models. Therefore, associated odds ratios should be interpreted cautiously, and will benefit from confirmation in future studies of ED patient populations with mTBI. As with all trauma studies, we were limited to patients who returned for follow-up, and attrition bias limits the generalizability of our study. Toxicology and blood alcohol level data were subject to institutional limitations and were not routinely drawn on all mTBI patients, which limits generalizability.

As the TRACK-TBI Pilot study variables were based on version 1 of the NINDS CDEs, insurance status, readmission, and granular post-discharge data were not available. Accordingly, post-acute interventions such as referrals and/or rehabilitation were not available. As we analyzed patients who were discharged from the ED, polytrauma was not categorized by the trauma registry, which is only available for admitted patients; however, this is likely a minor contribution, as patients’ peripheral injuries were deemed safe for discharge directly from the ED. Whether patients received TBI education materials is another data point informative for outcomes analysis [51] and was not tracked as part of the TRACK-TBI Pilot. These variables are being collected as part of the ongoing 18-center Transforming Research and Clinical Knowledge in Traumatic Brain Injury study [67] using version 2 of the NINDS TBI CDEs [68].

Regarding future directions, facilitating education and managing psychiatric issues may help reduce exposure and improve adherence to mTBI follow-up [51,52]. Comparison to military studies will also be of use, as the presence of psychiatric comorbidities, e.g., post-traumatic stress disorder (PTSD), consistently predicted poorer post-deployment outcome in deployed veterans compared to veteran controls, while demographics and injury mechanisms were less important [69,70,71]. This may be related to factors of intensity, and the frequency of trauma in military compared to civilian populations, as well as measures of resilience [72], which are ongoing topics of research. The implementation of qualified biomarkers sensitive to the presence of CT abnormalities can also improve diagnosis and the need to triage to follow-up for ED patients with mTBI [73,74].

## 5. Conclusions

In summary, it is our responsibility to implement compensatory resources and services for the over 65% of CT-negative mTBI patients who may predictably have functional deficits in the three to six months postinjury. The predictors of education, assault as the mechanism of injury, minority races, and the presence of baseline psychiatric disorders can be routinely collected at ED admission without significant burden to the provider or patient. These patients may benefit from education, follow-up resources, necessary referrals and/or early surveillance in clinic. Education, psychiatric history, race and ethnicity, along with mechanism of injury, should be considered for integration into current ED mTBI protocols.

## Figures and Tables

**Figure 1 brainsci-10-00269-f001:**
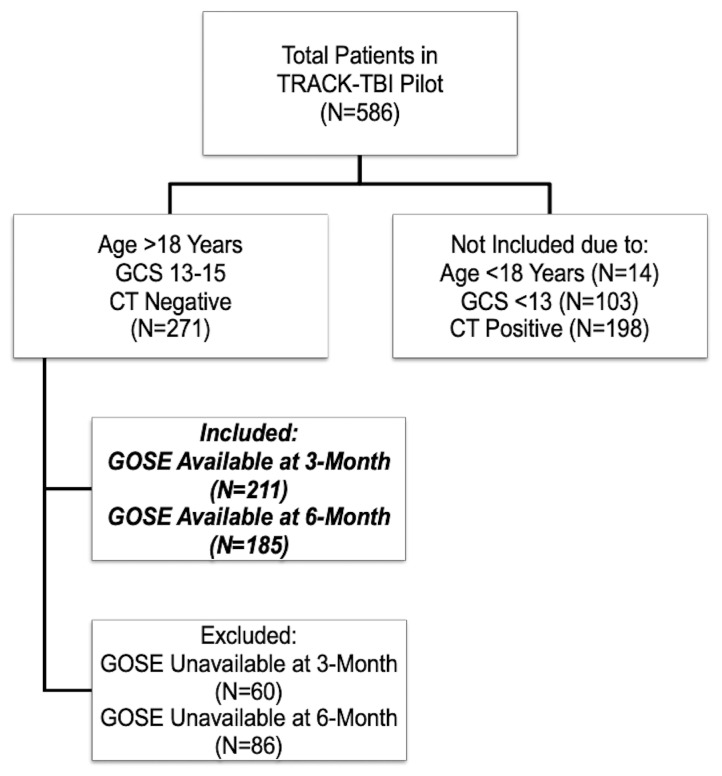
Flowchart of includedsSubjects.

**Table 1 brainsci-10-00269-t001:** Univariate associations between demographic and clinical characteristics and functional deficits at 3 and 6 months.

		3 Months, n = 211					6 Months, n = 185				
Variable		GOSE < 8 (n = 127)		GOSE = 8 (n = 84)		*p*-Value	GOSE < 8 (n = 121)		GOSE = 8 (n = 64)		*p*-Value
**Age**	Years (med, IQR)	40	26–54	36	26–50	0.30	40	26–52	35	24–51	0.37
**Gender**	Male	84	59%	59	41%	0.55	79	63%	46	37%	0.41
	Female	43	63%	25	37%		42	70%	18	30%	
**Race**	Caucasian	83	54%	72	46%	**0.02**	80	60%	54	40%	**0.07**
	African American or African	18	75%	6	25%		15	75%	5	25%	
	Asian	12	86%	2	14%		12	92%	1	8%	
	American Indian or Alaskan	2	100%	0	0%		2	100%	0	0%	
	Hawaiian or Pacific Islander	2	100%	0	0%		4	100%	0	0%	
	More than One Race	6	60%	4	40%		7	64%	4	36%	
	Not Available	1	100%	0	0%		1	100%	0	0%	
**Ethnicity**	Non-Hispanic	96	58%	70	42%	0.23	91	61%	57	39%	**0.03**
	Hispanic	31	69%	14	31%		30	81%	7	19%	
**Years of Education**	Years (med, IQR)	13	12–16	15	12.5–16	**0.006**	14	12–16	15	13–16.5	**0.002**
**Psychiatric History**	No	72	53%	63	47%	**<0.01**	63	56%	50	44%	**<0.001**
	Yes	55	72%	21	28%		58	81%	14	19%	
**Injury Mechanism**	Fall/Accident/Other	101	56%	78	44%	**0.01**	95	62%	59	38%	**0.02**
	Assault	26	81%	6	19%		26	84%	5	16%	
**Glasgow Coma Scale**	13	5	71%	2	29%	0.81	5	100%	0	0%	0.22
	14	28	62%	17	38%		24	60%	16	40%	
	15	94	59%	65	41%		92	66%	48	34%	
**Anticoagulant Use**	No	120	60%	79	40%	>0.99	112	65%	61	35%	0.55
	Yes	7	58%	5	42%		9	75%	3	25%	
**Loss of consciousness**	None	39	61%	25	39%	0.58	31	60%	21	40%	0.75
	< 30 min	54	57%	41	43%		59	69%	27	31%	
	≥ 30 min	4	50%	4	50%		7	64%	4	36%	
	Unknown	30	68%	14	32%		24	67%	12	33%	
**Post-traumatic amnesia**	None	48	61%	31	39%	0.66	45	63%	26	37%	0.61
	< 30 min	41	56%	32	44%		39	66%	13	34%	
	≥ 30 min	21	60%	14	40%		20	61%	13	39%	
	Unknown	17	71%	7	29%		17	77%	5	23%	
**Pre-hospital hypotension** **(SBP < 90)**	No	96	58%	69	42%	0.47	83	60%	56	40%	0.15
	Yes	6	75%	2	25%		7	88%	1	13%	
	Unknown	25	66%	13	34%		31	82%	7	18%	
**Pre-hospital hypoxia** **(SpO2 < 90)**	No	96	58%	70	42%	0.08	85	60%	56	40%	0.28
	Yes	5	100%	0	0%		3	100%	0	0%	
	Unknown	26	65%	14	35%		33	80%	8	20%	
**Urine toxicology screen**	Negative	120	61%	78	39%	0.77	113	65%	62	35%	0.5
	Positive	7	54%	6	46%		8	80%	2	20%	
**Blood alcohol level**	Negative	27	60%	18	40%	0.66	23	62%	14	38%	0.41
	Positive	16	70%	7	30%		18	78%	5	22%	
	Not obtained	84	59%	59	41%		80	64%	45	36%	
**ED hypotension**	No	125	60%	83	40%	>0.99	119	65%	64	35%	0.55
	Yes	1	50%	1	50%		2	100%	0	0%	
	Unknown	1	100%	0	0%		0	0%	0	0%	
**ED hypoxia**	No	120	60%	81	40%	>0.99	116	65%	62	35%	>0.99
	Yes	5	63%	3	38%		4	67%	2	33%	
	Unknown	2	100%	0	0%		1	100%	0	0%	

Caption: ED—emergency department; IQR—interquartile range; GOSE—Glasgow Outcome Scale—Extended; SBP—systolic blood pressure; SpO_2_—oxygen saturation measured by pulse oximetry. Bold font designates *p* < 0.05.

**Table 2 brainsci-10-00269-t002:** Multivariable analysis of Risk Factors for 3- and 6-month functional deficits (GOSE < 8).

Risk Factors	GOSE < 8 vs. GOSE = 8 at 3 Months Postinjury		GOSE < 8 vs. GOSE = 8 at 6 Months Postinjury	
	AOR (95% CI)	*P*-Value	AOR (95% CI)	*P*-Value
Years of Education	0.88 (0.78–0.998)	0.046	0.85 (0.74–0.98)	0.026
Psychiatric History				
No	1	0.015	1	0.001
Yes	2.30 (1.18–4.49)		3.75 (1.73–8.12)	
Injury Mechanism				
Fall/Accident/Other	1	0.025	1	0.054
Assault	3.53 (1.17–10.63)		3.41 (0.98–11.85)	
Race				
White	1	0.006	1	0.025
Black	2.53 (0.82–7.83)		1.77 (0.48–6.51)	
Asian/Other	12.40 (2.66–57.77)		23.99 (2.93–196.84)	
More than one race	0.88 (0.20–3.84)		1.03 (0.22–4.70)	
Ethnicity				
Non-Hispanic	1	0.124	1	0.014
Hispanic	1.84 (0.85–4.02)		3.48 (1.29–9.37)	

Caption: GOSE—Glasgow Outcome Scale—Extended; AOR—adjusted odds ratio.

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
