# Peer review of "Clinical Predictors of 3- and 6-Month Outcome for Mild Traumatic Brain Injury Patients with a Negative Head CT Scan in the Emergency Department: A TRACK-TBI Pilot Study"

_brainsci, 2020, doi:10.3390/brainsci10050269_

Round 1
Reviewer 1 Report
This study uses the data from the TRACK-TBI Pilot study, a multi-site study conducted at three level I trauma centers in the United States. The main goal is to assess a cohort of patients in the ED with mTBI who are at greater risk for adverse 76 neurocognitive sequelae despite an initial negative head CT scan, and assess whether traditional 77 clinical factors measured in the ED in this patient population are adequate for predicting outcome at 78 3- and 6-months postinjury.
I do have a concern that many subgroups have small numbers such as Asian. Therefore the confidence intervals for the AOR is so wide, which suggests the results are not reliable. Authors acknowledge this limitation but fail to elaborate implications for interpreting their results. I have a feeling that authors reach a more confirmative conclusions than what should be based on the results.
My second concern is that authors do not find association between GSC scores and outcomes. This may be due to small sample size and very narrow range of the GSC scores.
Overall, the study design is fine and the contribution to the field is minor to moderate.
Author Response
Reviewer 1:
I do have a concern that many subgroups have small numbers such as Asian. Therefore the confidence intervals for the AOR is so wide, which suggests the results are not reliable. Authors acknowledge this limitation but fail to elaborate implications for interpreting their results. I have a feeling that authors reach a more confirmative conclusions than what should be based on the results.
We thank the Reviewer for this point, which is unfortunately a limitation of studies with smaller sample sizes and unequal race distributions. We have added the following in the limitations: “The number of participants in non-White racial groups is small. Therefore, associated odds ratios should be interpreted cautiously, and will benefit from confirmation in future studies of ED patient populations with mTBI.”
My second concern is that authors do not find association between GSC scores and outcomes. This may be due to small sample size and very narrow range of the GSC scores.
Agree with reviewer’s assessment. This study focuses on mild TBI and therefore only patients with GCS 13-15 were included. As the GCS variability was not large, and the vast majority of patients (>95%) had GCS 14-15, it was not surprising that GCS was not a predictor of outcomes. This has been found in prior studies (Jacobs et al. J Neurotrauma 2010. PMID 20035619).

Reviewer 2 Report
Multicenter TRACK-TBI provides an excellent venue for discovery of pre- and post-hospital factors that contribute to disparities in mild TBI outcomes. The heterogeneity in TBI itself is confounding; factoring these additional parameters to improve patient outcomes is daunting task. The authors have tackled this issue very well and made good use of their access to TRACK-TBI data. The study population is mild traumatic brain injury (mTBI) patients defined as (Glasgow Coma Scale 13-15) and negative ED initial head CT <24h of injury that fail to return to baseline 3-months or more post injury. The study seeks to identify outcome predictors using standard methods with outcome as tool to dichotomize data. The study identified less education, prior psychiatric history, ‘Asian, Hispanic ethnicity’ compared to White as risk factors for 6-month GOSE<8. Consistent with literature assault predicts poorer outcome. Together these factors prolong disability and targeting these could help mitigate disparity.
Please make an effort to compare the data with those from military mild TBI. It may then help uncover the role physical fitness, role of race. Given that race is immutable. Facilitating education, managing psychiatric issues may help reduce exposure as well as improve outcomes in case of incidence. Factors such as economic status, non-uniform blood alcohol levels in data set, small cohort of White patients do limit generalizability. Use of a sensitive biomarker to normalize injury could help appearance of racial differences for injury classified as assault.
Author Response
Reviewer 2:
Multicenter TRACK-TBI provides an excellent venue for discovery of pre- and post-hospital factors that contribute to disparities in mild TBI outcomes. The heterogeneity in TBI itself is confounding; factoring these additional parameters to improve patient outcomes is daunting task. The authors have tackled this issue very well and made good use of their access to TRACK-TBI data. The study population is mild traumatic brain injury (mTBI) patients defined as (Glasgow Coma Scale 13-15) and negative ED initial head CT <24h of injury that fail to return to baseline 3-months or more post injury. The study seeks to identify outcome predictors using standard methods with outcome as tool to dichotomize data. The study identified less education, prior psychiatric history, ‘Asian, Hispanic ethnicity’ compared to White as risk factors for 6-month GOSE<8. Consistent with literature assault predicts poorer outcome. Together these factors prolong disability and targeting these could help mitigate disparity.
We thank the Reviewer for their time and support of our work.
Please make an effort to compare the data with those from military mild TBI. It may then help uncover the role physical fitness, role of race. Given that race is immutable. Facilitating education, managing psychiatric issues may help reduce exposure as well as improve outcomes in case of incidence. Factors such as economic status, non-uniform blood alcohol levels in data set, small cohort of White patients do limit generalizability. Use of a sensitive biomarker to normalize injury could help appearance of racial differences for injury classified as assault.
We thank the Reviewer for this important point, and have added the following to the Limitations section regarding comparison to military studies:
“Comparison to military studies will also be of use, as presence of psychiatric comorbidities, e.g. post-traumatic stress disorder (PTSD), consistently predicted poorer post-deployment outcome in deployed veterans compared to veteran controls, while demographics and injury mechanisms were less important.(68,69,70) This may be related to factors of intensity, and frequency of trauma in military compared to civilian populations, as well as measures of resilience,(71) which are ongoing topics of research.”
We have included the following addition in the Limitations as suggested by the Reviewer:
“The number of participants in non-White racial groups is small. Therefore, associated odds ratios should be interpreted cautiously, and will benefit from confirmation in future studies of ED patient populations with mTBI. As with all trauma studies, we were limited to patients who returned for follow-up and attrition bias limits the generalizability of our study. Toxicology and blood alcohol level data were subject to institutional limitations and were not routinely drawn on all mTBI patients, which limits generalizability.”
“Regarding future directions, facilitating education and managing psychiatric issues may help reduce exposure and improve adherence to mTBI follow-up.(48,49) The implementation of qualified biomarkers sensitive to presence of CT abnormalities can also improve diagnosis and need to triage to follow-up for ED patients with mTBI.(65, 66)”

Round 2
Reviewer 1 Report
Although authors responded to my two major concerns, I am still not very satisfied with the response. Due to the inherent limitation in their data, these two concerns are hard tp address.
Overall, the contribution of the manuscript to the field is limited because of the major limitations.